# Uncharged Monolithic Carbon Fibers Are More Sensitive to Cross-Junction Compression than Charged

**DOI:** 10.3390/s24123937

**Published:** 2024-06-18

**Authors:** Oleksandr Syzoniuk, Saoni Banerji, Alvo Aabloo, Indrek Must

**Affiliations:** IMS Lab, Institute of Technology, University of Tartu, Nooruse 1, 50411 Tartu, Estonia

**Keywords:** activated carbon cloth electrodes, electrical double layer, touch contacts, pressure sensors, charge sensors, smart wearables

## Abstract

Textile-based wearable robotics increasingly integrates sensing and energy materials to enhance functionality, particularly in physiological monitoring, demanding higher-performing and abundant robotic textiles. Among the alternatives, activated carbon cloth stands out due to its monolithic nature and high specific surface area, enabling uninterrupted electron transfer and energy storage capability in the electrical double layer, respectively. Yet, the potential of monolithic activated carbon cloth electrodes (MACCEs) in wearables still needs to be explored, particularly in sensing and energy storage. MACCE conductance increased by 29% when saturated with Na_2_SO_4_ aqueous electrolyte and charged from 0 to 0.375 V. MACCE was validated for measuring pressure up to 28 kPa at all assessed charge levels. Electrode sensitivity to compression decreased by 30% at the highest potential due to repulsive forces between like charges in electrical double layers at the MACCE surface, counteracting compression. MACCE’s controllable sensitivity decrease can be beneficial for garments in avoiding irrelevant signals and focusing on essential health changes. A MACCE charge-dependent sensitivity provides a method for assessing local electrode charge. Our study highlights controlled charging and electrolyte interactions in MACCE for multifunctional roles, including energy transmission and pressure detection, in smart wearables.

## 1. Introduction

Wearable robotics, an emerging field at the intersection of biomechanics and engineering, enhances human mobility and task execution in areas like rehabilitation and occupational health using a variety of mechanisms, including power-assisted actuators and passive structures [1,2]. Integral to the development of wearable robotics are advancements in textile technologies, such as conductive fabrics or stretchable electronics [3,4]. The selection of materials is therefore guided not only by their inherent properties, such as tensile strength, thermal stability, and elasticity, but also by the materials’ ability to meet specific operational demands, including conductance, sensitivity, and mechanical flexibility [5]. These demands are critical for wearable applications, encompassing sensitivity for physiological signal capture [6,7] and other textile-based systems involving intermediary energy storage [8]. Moreover, these applications must sustain efficiency and ensure reliable signal transmission, even under varying environmental conditions [9]. This approach to material selection ensures that the materials’ performance aligns with the goals of wearable robotics, encompassing energy storage, signal transmission, and user interaction [10].

As wearable robotics evolves, materials become essential to meet functional demands. Materials with mechanical integrity and electrical conductivity ensure robust and responsive wearable devices. However, beyond these general requirements, certain wearable robotics solutions demand materials with a high specific surface area (SSA). For instance, in capacitive energy storage systems, high SSA is essential for increasing capacitance by allowing more ions (from an electrolytic medium) to accumulate on the surface of the electrodes of a device. Moreover, in textile actuators based on an electric double layer (EDL), the actuation magnitude is proportional to the high SSA [11]. Among the potential candidates, activated carbon cloth (ACC) stands out as a promising option. Its high SSA, typically between 500 to 1200 m^2^/g [12], enables a high ion adsorption capacity, crucial for textile-based intermediary energy storage components [13,14] and also beneficial for textile actuators [15], thus positioning ACC as the preferred material for these applications, compared to traditional, non-porous materials (copper wire, carbon fiber, etc.). While other materials (such as graphene oxide, conductive polymers, etc.) may exhibit high SSAs, ACC’s set of properties, extending beyond mere energy storage, sets it apart. Unlike other high SSA carbon materials (e.g., carbon black), ACC is not limited to composite forms and can be utilized in meter-scale monolithic structures due to its monolithic nature. The monolithic structure of ACC enhances the efficiency of wearable intermediary energy storage components, promoting efficient electron transfer and reducing system impedance upon charging (reducing the electron tunneling effect and optimizing the electron path) [16]. Additionally, the electrohydrodynamic actuators and intermediary capacitive energy storage components share the optimization targets of high SSA and low impedance for the electrodes, thus motivating their concurrent or intermittent use for both functions. ACC electrodes act as electron-to-ion transducers [17]. When an electric charge is applied to the electrodes, the ions in the electrolyte migrate within the whole ion-permeable structure, including a porous polymer matrix. The number of displaced ions per applied potential, and thus actuation and stored energy, is proportional to the specific surface area of the electrodes. Ion migration, driven by the electric field between the electrodes, leads to non-uniform swelling in case the mobilities of cations and anions are not equal (achieved typically via other components rather than electrodes), whereas EDLCs favor high mobility for all charge carriers. Although the mobility equality target is different for actuation and energy storage, we believe the dual-function use in a wearable scenario is practical because of volume constraints. Previous research in the use of MACCE in water capacitive desalination supports the combination of ion displacement and energy storage, as some of the energy spent for ion adsorption could be recovered [18].

Although traditionally used in filtration systems for their adsorption capabilities, ACC’s applicability broadens significantly in wearable technologies. However, the roles of monolithic ACC electrodes in wearable technology, particularly in energy storage, actuation, and sensing, have not been fully explored, highlighting a key area for future research.

As the first aim, we aim to investigate the electrical behavior of touch contacts between bundles of monolithic fibers in MACCE under compression pressure (as depicted in Figure 1A,B). We anticipate that the system’s sensitivity will not be solely dependent on the number of touch contacts; it is also expected that the sensitivity response will be affected by tunneling paths between fibers in close proximity. The tunneling junctions are anticipated to introduce a non-linear responsiveness to changes in distance, making the system sensitive to even slight shifts in proximity between fibers [19,20,21]. We aim to identify the pressure sensitivity region of MACCE, which is influenced by a combination of factors, including changes in the number of touch contacts and the presence of tunneling junctions within the electrode under compression. These insights are expected to influence energy storage efficiency and signal transmission reliability. For instance, for textile capacitive components serving as intermediary energy storage, the reduced resistance at non-monolithic junctions due to compression could enhance its charging rate, making energy transfer and storage more efficient [22,23].

As the second aim, we assess how liquid media influences MACCE’s conductance at non-monolithic junctions and analyze the roles of liquid-induced capillary forces and the electric double layer (as visualized in Figure 1C). The high SSA actuators and energy storage components, as those discussed above, typically operate in liquid immersion or in wet environments. We are interested in the dynamics when MACCE is exposed to an electrolytic solution imitating the influence of moisture or sweat, a common scenario in wearable robotics. In this context, liquid-induced capillary forces may impact the physical proximity of conductive elements, thereby affecting conductance [24,25]. Concurrently, the presence of an electric double layer at the interface between the liquid and the conductive material introduces a dual-pathway conductance mechanism, where ionic transport through the solution and electronic transport through the conductive network operate in tandem, enhancing the conductive properties of MACCE [16].

The third aim of our research is to evaluate the physical properties of MACCE as a sensing platform for wearable sensors. Specifically, we examine the sensitivity and response range of MACCE subjected to varying levels of compression pressure. We expect the MACCE to display characteristics similar to known force-sensitive resistors, which operate through the interaction of conductive layers under pressure. However, unlike membrane switches that require pressure to establish an electrical connection, MACCE maintains a continuous conductive state that is improved, rather than initiated, by compression. Overall pressure sensitivity and conductivity of textile MACCE could be advantageous for detecting and monitoring physiological states such as respiration and cardiac activity [26,27].

The paper is organized as follows: Section 2 outlines the fabrication and characterization of MACCE. Section 3 analyzes how the conductance of MACCE responds to varying pressure and electrolyte conditions. This section also demonstrates MACCE’s conductance response to compression in a pre-charged state. Section 4 concludes with key insights into the conductance and sensitivity behavior of MACCE and its potential applications.

## 2. Experimental

### 2.1. Fabrication of MACCE System

#### Electrode Fabrication

Electrodes (Figure 2) were fabricated by cutting a 100 × 100 mm piece of ACC (Zorflex FM10 by Calgon Carbon). Copper wires were attached to establish electrical connections for resistance measurements of the MACCE: one along the length (warp direction) and the other across the width (weft direction). The connection between MACCE and copper wires was reinforced with silver-based conductive ink (Electon 40AC). All ACC fibers in each direction were interconnected, focusing on the electrical characteristics determined by distributed non-monolithic junctions between warp and weft threads. The wire connection areas were encapsulated using tin-cured silicone (POLASTOSIL M-2000) to prevent reactions with the electrolyte. The tin-cured silicone prevented the capture of curing catalyzers from the uncured silicone by MACCE.

### 2.2. Electromechanical Characterization of MACCE

The frequency-independent resistance of MACCE was characterized in a two-electrode setup depicted in Figure 3A. Two wires from a single MACCE were connected to the Biologic BP300 impedance analyzer in a two-electrode configuration and in counter-electrode-to-ground mode. The resistance measurements were conducted at 100 kHz frequency at the open-circuit voltage and 10 mV amplitude. Conductance was calculated as the reciprocal of resistance.

#### 2.2.1. Pressure Sensitivity

Varying compression levels were induced by first placing MACCE on a horizontal flat non-conductive surface and then sequentially placing steel balance weights with a cylindrical non-conductive base with an interfacial contact area of 6.16 cm^2^ (radius = 1.4 cm) on top of the MACCE. The weights ranged from 0.01 kg to 1.3 kg. Each cylindrical weight was manually centered on the MACCE’s pressure-sensitive area. Following each pressure application, sixty consecutive resistance readings were taken. The electrode sensitivity and dynamic range were determined through fitting the force–pressure relationship to a linear or exponential model using Python Scipy library.

#### 2.2.2. Measurements in Electrolytic Media

The MACCE electrode was saturated with a 30 mM Na_2_SO_4_ (Sigma Aldrich, St. Louis, MO, USA, 99%, analytical grade) aqueous solution by manually dipping the electrode. Sixty baseline resistance readings were taken post-saturation.

#### 2.2.3. MACCE Charging

The EDL on MACCE was charged in a two-electrode setup. A stack of the sensing MACCE and an additional MACCE, separated by a 0.15 mm thick filter paper to prevent contact between MACCEs, was assembled. The additional MACCE had a wire connection only on one side.

The two-electrode system was connected to a power supply, as shown in Figure 3B. The impedance analyzer and the power supply were connected to share a common ground point.

First, the power supply charged the system to the chosen voltage level for 30 min. The MACCE was then disconnected from the power source for MACCE’s conductance and pressure sensitivity measurements. The system was charged from 0 to 0.75 V, corresponding to MACCE potentials 0 to 0.375 V in the mirror-symmetric system.

## 3. Results and Discussion

### 3.1. Pressure-Dependent Conductance in MACCE

Figure 4 shows the time-domain conductance (G) variation in MACCE under dynamically applied pressure. Conductance values were recorded with the intermittent application of a 3 kPa pressure at 50 s intervals, marked by the gray shaded areas.

When the pressure was applied, the conductance consistently increased from the baseline at 0.0574 S to approximately 0.058 S. The indicated conductance enhancement of roughly 0.00065 S or about 1% from the baseline, highlighting MACCE’s ability to respond to relatively low-pressure levels. The increase in conductance resulted from an increased number of touch contacts between conductive fibers and reduced tunneling distances, facilitating better electron transfer. After pressure was removed, conductance returned to baseline values, affirming the reversible behavior of the electrode under cyclic loading conditions.

### 3.2. Conductance Enhancement in MACCE through Electrolyte Interaction

Figure 5 compares the trends of relative conductance changes with respect to the applied pressure in cases of electrolyte-infused and dry MACCE. The wet system demonstrated a linear pressure-sensitive region extending up to 15.5 kPa, with a sensitivity of 2.38 MPa^−1^. In contrast, the dry system exhibited an exponential decay pattern, characterized by an initial sensitivity of 3.6 MPa^−1^ at 0 kPa, which did not develop a linear range. In the dry state, the saturation of physical contacts, with the potential additional contribution of tunneling currents exponential to cross-fiber distance, expectedly provided a nonlinear response. Pressures above 15.5 kPa did not increase the conductance for either case, corresponding to conductance variation ranges of approximately 4% and 2%, respectively.

The conductance enhancement in the wet case is attributed to the liquid electrolyte promoting capillary action, which draws fibers closer together, increasing touch contacts, reducing the distance for ion transport, and providing additional pathways for ions to move. As these fibers come closer under pressure, the capillary forces are heightened due to the reduction in the cross-fiber distance and an increase in the curvature of the meniscus formed by the liquid bridges [28]. Cohesive capillary forces between fibers result in a variable-distance conduction pathway, additional compared to the dry case, between the adjacent and intersecting fibers. Multiple concurrent mechanisms in the wet case contribute to a larger range and linear response. The absence of a linear range in the dry case highlights the positive effect of liquid media on electrode sensitivity, particularly in case of MACCE’s utilization as a pressure sensor.

### 3.3. Effect of Charge on MACCE Pressure-Sensitivity

Figure 6 illustrates the pressure-dependent conductance response of MACCE depending on the charging extent of the electrical double layer at its surface.

We observed a linear increase in conductance in MACCE up to 15.5 kPa across all charging levels, from 0 V to 0.375 V. The uncharged electrode provided the largest sensitivity of 0.152 µS/Pa (conductance variation of 29%), diminishing to 0.046 µS/Pa at the largest observed electrode potential of 0.375 V. When pressure was applied, the pressure sensitivity decreased at an increasing charging extent. In contrast, the increased potential amplified this repulsive interaction, leading to a pronounced decrease in conductance value.

Figure 7 compares the conductance and sensitivity trends. Although the charged fibers have more space to approach each other when pressed together, the applied pressure unexpectedly did not increase sensitivity, possibly because of the additional degree of freedom of lateral fiber displacement.

The observed decrease in the pressure-sensitivity of conductance at an increasing charge can be explained as follows. Upon charging the electrode, EDLs form along the longitudinal and across fibers of MACCE, creating local EDL overlaps. The increase in conductance at increased charging levels is attributable to the monolithic nature of MACCE, which allows for uninterrupted electron flow across the electrode, thereby supporting efficient electron transfer [16]. Additionally, the formation of an electrical double layer at the electrode surfaces with charging enhances the local charge density, further boosting (absolute) electrical conductance.

The reduction in pressure sensitivity appears due to increased repulsive forces among like charges within the electric double layers of MACCE fibers. In concentrated electrolyte solutions, EDL forces are short-range, confined to approximately 0.5 nm [29]. When electric double layers are strongly overlapping, the disjoining pressure profile exhibits an algebraic decay, affecting particle behavior at close proximity. Repulsive disjoining pressures can reach up to 10^5^ Pa at separations up to 5 nm [30]. As potential increases, these forces intensify and counteract fiber compression, thus decreasing the sensitivity to mechanical stimuli. The blue diamond in Figure 7 shows the baseline conductance of MACCE in its dry state, represented by a single reading at zero voltage, showing how the conductance is increased by introducing an electrolyte, which elevates the baseline conductance by 10%. Therefore, as the more charged electrode has increased conductance and lower sensitivity to pressure, it performs better as a signal and power transmitter. However, the proportional increase is rather small in magnitude; in practical configurations, it only moderately influences power delivery kinetics. Vice versa, the lowest charging extent provides the best sensing. Conversely, the charge-dependent sensitivity can provide a valuable practical method to assess the local electrode charging extent!

Due to the fragility of the ACC fibers exposed to flexure, the MACCEs are hardly competitive solely as pressure sensors. However, the pressure-sensitivity is a valuable feature of the MACCE as a multifunctional system component. The observed charge-dependent sensitivity of MACCE offers a method for measuring the local charge state within convoluted systems. This feature is especially valuable for capacitive components constructed from MACCE, which applies to wearable robotics. By monitoring changes in sensitivity, we can directly infer the charge status when local charging state assessment by voltage differential is not available, thereby enhancing the operational efficiency and versatility of wearable devices designed for intermittent energy storage.

### 3.4. Pressure Sensitivity and Conductance Dynamics in MACCE

Figure 8 illustrates the conductance (G) response of a dry MACCE when subjected to compressive pressures ranging from 0 to 28 kPa at a higher resolution. A relatively small insensitive region was observed up to 1.8 kPa, suggesting that the applied pressure was not sufficient to decrease fiber spacing enough to create new touch contacts to alter MACCE’s conductance. Moreover, as discussed above, tunneling junctions were expected to be highly sensitive to minor pressure variations and vary electrode conductance, thus not predicting pressure-insensitive areas at small pressures. A possible reason for this behavior could be that the applied load is not transmitting to the fiber junctions in the woven textile, even slightly, at low pressures up to 1.8 kPa. This lack of expected transition might be influenced by other yet-to-be-understood aspects of the electrode’s structure. Further detailed studies are necessary to explore these components and their impact on the electrode’s behavior.

The continuous pressure sensitivity, without observable discrete conductance levels, provides evidence of tunneling effects, as the progressive narrowing gaps between closely situated fibers affect conductance not only by the number of touch contacts but also by changing tunneling distances.

## 4. Conclusions

Our study of MACCE highlights its potential in advancing wearable technologies as a single-material framework for sensing and energy transmission. This multifunctionality of MACCE is advantageous for several reasons. First, the electrode can serve both as a sensing platform and an energy transmitter. Consequently, MACCE can dynamically respond to environmental changes and user interactions. It can contribute to garments that actively monitor physiological signals. We observed a 29% increase in conductance upon electrode charging. This observation confirms that MACCE has charge-controlled functionality at the material level. Moreover, the MACCE intrinsic controllability feature can be tapped to integrate efficient energy management systems into smart wearables. Second, the study also uncovers a decrease in electrode sensitivity at higher charge densities. This characteristic enables the precise control of signal capture, which is essential for diminishing background noise and focusing on vital health indicators in wearable health monitoring systems.

The charge-dependent sensitivity of MACCE provides a method to assess the local charge state within the electrode and offers new directions in research. This MACCE charge-controlled feature opens more possibilities for developing diagnostic tools to monitor and manage energy dynamically in wearable technologies. Each of these findings underscores the multifaceted applications of MACCE, making it a strategic component for the ongoing evolution of wearable devices.

## Figures and Tables

**Figure 1 sensors-24-03937-f001:**
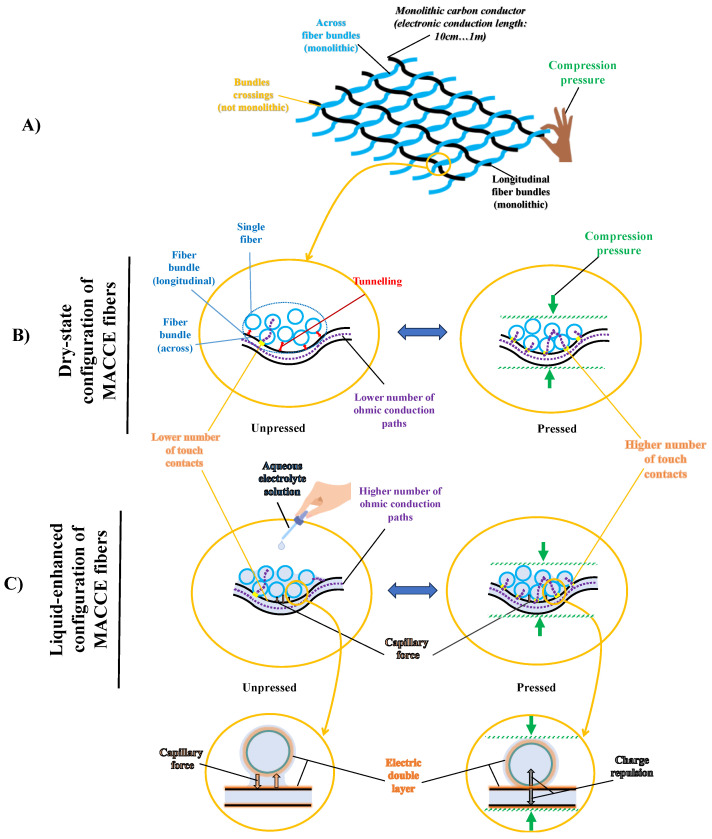
MACCE pressure sensitivity in dry and wet environments. (**A**) MACCE’s structure: arrangement of monolithic longitudinal and cross-fiber bundles with applied compression pressure. (**B**) Dry-state electronic conduction of MACCE, affected by the number of touch contacts and tunneling gap width. (**C**) Wet electronic and ionic conduction of MACCE, affected by capillary convergence of adjacent fibers and repulsion between like charges in overlapping electric double layers at the fiber interfaces.

**Figure 2 sensors-24-03937-f002:**
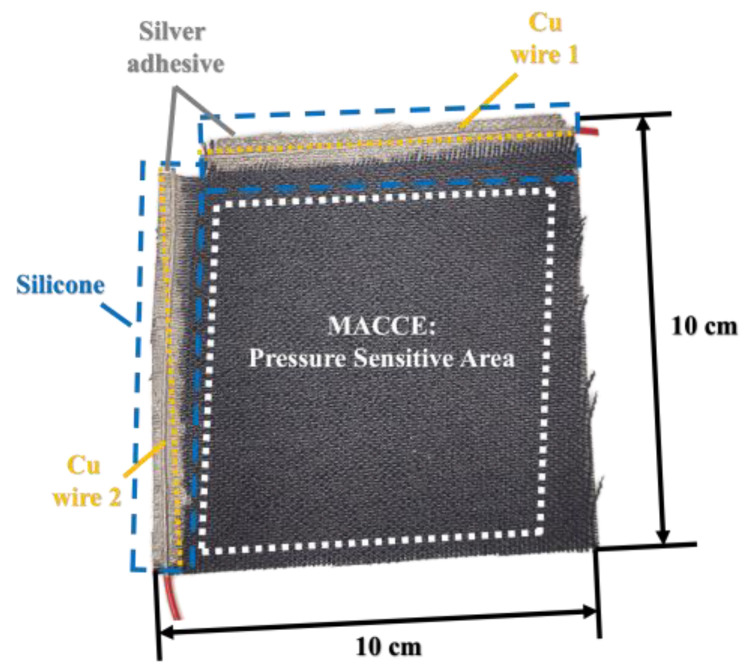
MACCE configuration with copper wire connections. White dashed line indicates the pressure-sensitive area.

**Figure 3 sensors-24-03937-f003:**
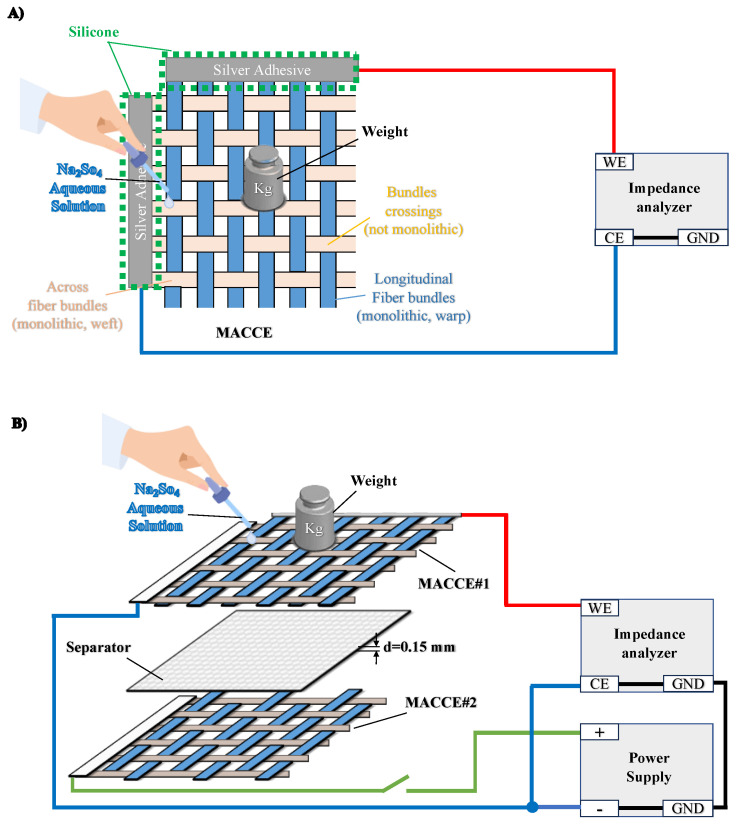
Experimental setup. (**A**) Measurement of pressure-dependent resistance of MACCE. (**B**) A two-electrode setup for charging the MACCE.

**Figure 4 sensors-24-03937-f004:**
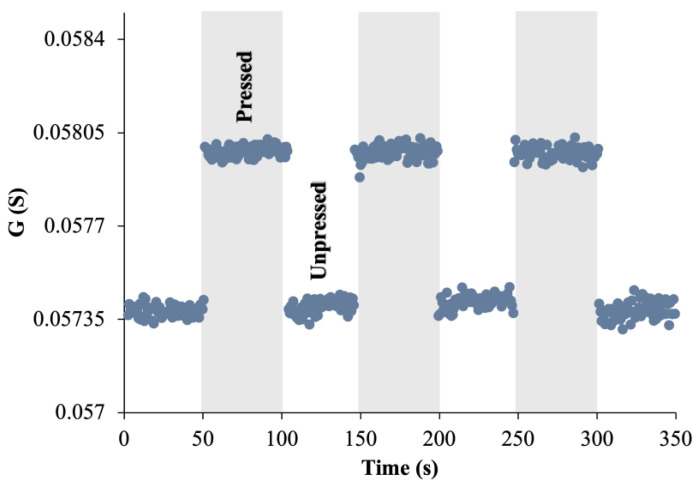
Dynamic conductance response of MACCE under intermittently applied 3 kPa pressure.

**Figure 5 sensors-24-03937-f005:**
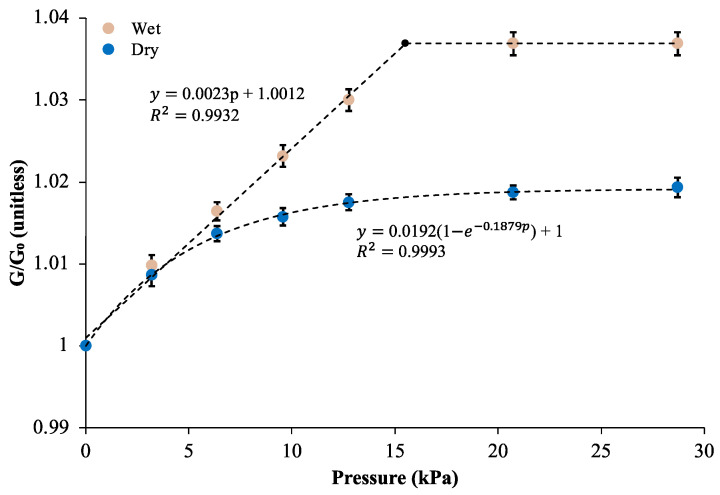
Pressure-induced conductance variation in MACCE in electrolyte-enhanced and dry conditions. Error bars represent standard deviation. The dashed line for the wet system illustrates linear fit. For the dry system, the dashed line indicates exponential fit.

**Figure 6 sensors-24-03937-f006:**
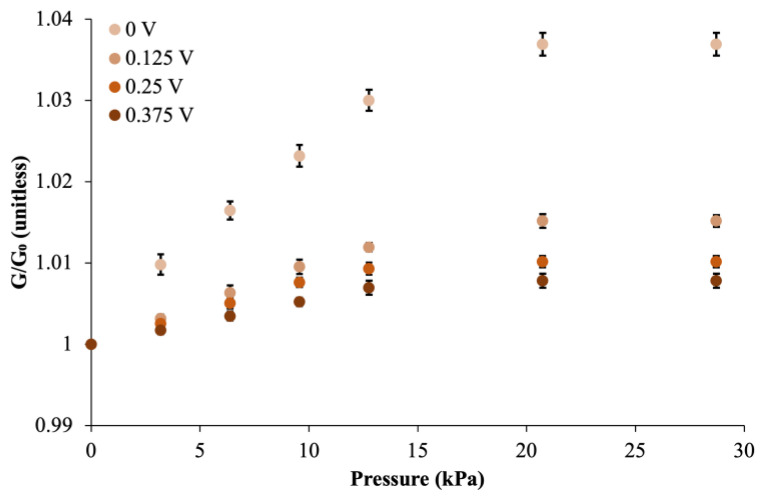
Normalized conductance at varying pressures: MACCE’s response to incremental applied pressures under different potentials. Error bars are standard deviation.

**Figure 7 sensors-24-03937-f007:**
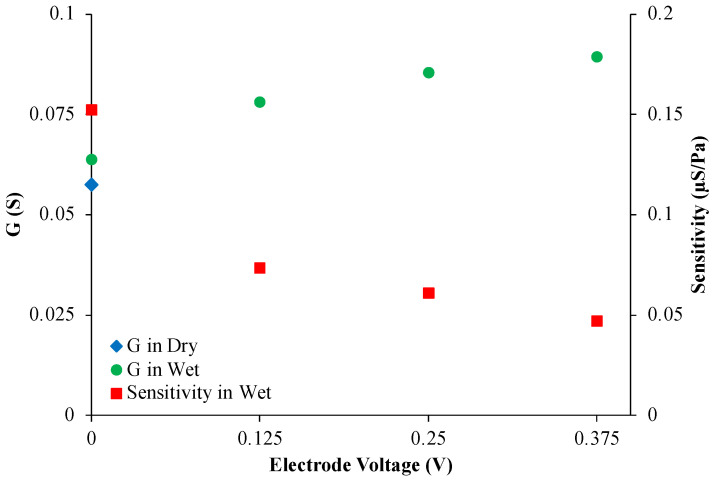
Conductance and sensitivity variation at different MACCE potentials. The blue diamond represents the conductance of uncharged dry MACCE.

**Figure 8 sensors-24-03937-f008:**
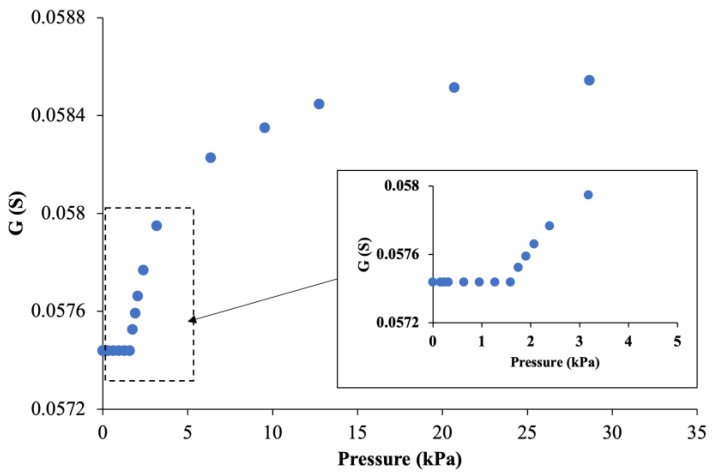
The pressure-insensitive region at low-pressure levels.

## Data Availability

All data that support the findings of this study are included within the article.

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
