# Peer review of "Uncharged Monolithic Carbon Fibers Are More Sensitive to Cross-Junction Compression than Charged"

_sensors, 2024, doi:10.3390/s24123937_

Round 1

Reviewer 1 Report

Comments and Suggestions for Authors

This paper systematically investigated pressure-dependent conductance in monolithic activated carbon cloth electrodes (MACCE) in the dry state or the wet state saturated with when saturated with Na2SO4. The article was well-written and organized. The authors observed an increase in conductance in MACCE until saturation, and the electrolytic media resulted in a 7% increase in pressure sensitivity, while this pressure sensitivity decreases with the increased voltage across the electrode. These findings offer new directions in research.

I have several suggestions and questions as stated following:

1.     Title: the title is not easy to understand and is not accurate. You should state “less sensitive” compared to “what.”

2.     On Page 6, Lines 206-207 and Figure 5. I am a bit worried about the fitting for the dry-state data. The data points at 10 and 12.5kPa are deviated from the fitting line. A curve-fitting line might be better. Could you provide any reference or mechanism that supports this linear relationship?

3.     On Page 7, Lines 222-223. I agree with you that capillary force increases conductance. How does a rise in capillary force increase pressure sensitivity? I suggest you provide an explanation of how pressure influences capillary force.

4.     On Page 8, Lines 240-242 and Lines 257-258. Electric double-layer force is a long-range weak force. You state that the EDL force can counteract the compression force. Can you provide some calculations that support this explanation? Or could you give some reference support?

Reviewer 2 Report

Comments and Suggestions for Authors

I found your paper interesting and well-written. I would  comment that the first use of G comes (Fig 4) well before you define it as conductance in Section 3.4 (while it may be obvious, normally the definition of a symbol comes at first use). 

Another  omission is the method or procedure by which you compute conductance. You measure resistance so the essential process may be elementary, but as an archival paper, the details of how (and why - see below) are important.

Finally, given that an application of your work is in wearables, which are most often energized (as I understand) not by AC but by DC (a battery), some explanation as to why you chose to measure impedance instead of resistance should be given.

Reviewer 3 Report

Comments and Suggestions for Authors

This manuscript reports the characteristics of monolithic activated carbon cloth electrodes (MACCE) in different conditions. The authors illustrate the pressure sensitivity and charging properties and compare MACCE’s performance in dry and wet conditions. While the reviewer appreciates the author’s efforts, the following aspects should be polished prior to publication.

Major concerns:

1.      In both the introduction and conclusion part of this manuscript, the authors mention the role of MACCE in energy storage, actuation, and sensing. However, the focus of this literature is on the sensitivity performance and there is only limited content concerning the actuation. In this case, the reviewer suggests the authors either revise the relevant expression in the manuscript or add more information about the role of MACCE in terms of energy storage and actuation.

2.      In Figure 5 the data is not adequate to support the linear trend of pressure-sensitive region and there are no data in the range between 12 kPa and 22 kPa. Incorporating additional data is believed to enhance the experimental analysis, making it more comprehensive and logically robust.

Minor concerns:

3.      In Figure 7, there are no blue circles indicating the baseline conductance of MACCE in its dry state. The relevant content in the analyzing part should be further checked.
